# Multiple Magma Conduits Model of the Jinchuan Ni-Cu-(PGE) Deposit, Northwestern China: Constraints from the Geochemistry of Platinum-Group Elements

**Xiancheng Mao [1]** , **Longjiao Li [1]**, **Zhankun Liu [1,]\***, **Renyu Zeng [1,2]**, **Jeffrey M. Dick [1]**, **Bin Yue [3]** **and Qixing Ai [3]**

[1]   Key Laboratory of Metallogenic Prediction of Nonferrous Metals and Geological Environment Monitoring (Ministry of Education), School of Geosciences and Info-Physics, Central South University, Changsha 410083, China; mxc@csu.edu.cn (X.M.); lilongjiao@csu.edu.cn (L.L.); zengrenyu@126.com (R.Z.); jeff@chnosz.net (J.M.D.)

[2]   State Key Laboratory of Nuclear Resources and Environment, East China University of Technology, Nanchang 330013, China

[3]   Nickel Cobalt Research and Design Institute, Jinchuan Group Co., Ltd., Jinchang 737104, China; yuebin@sina.com (B.Y.); aiqixing@jnmc.com (Q.A.)

\*   Correspondence: zkliu0322@csu.edu.cn; Tel.: +86-731-888-77571

**Abstract:** The giant Jinchuan nickel-copper-platinum-group element (PGE) deposit is hosted by two individual sub-vertical intrusions, referred to as the western and eastern intrusions (including segment II-W and segment II-E). Exactly how the Jinchuan deposit was formed by a system of sub-vertical magma conduits is still not well understood. This paper reports new major elements, trace elements and PGEs data from the Jinchuan deposit to study the formation mechanism of sulfide ores with different textures and their relationship with the magma conduit system. Our study shows that the PGE tenors of disseminated and net-textured sulfide in segment II-E are significantly lower than segment II-W and the western intrusion, but the Cu/Pd ratios are opposite. In addition, net-textured sulfides in segment II-W show a negative correlation between IPGE (Ir, Ru and Rh) and PPGE (Pt and Pd) in contrast to the positive correlation in segment II-E and the western intrusion. These features indicate the parental magma sources of the western intrusion, segment II-W and segment II-E were originally three different surges of PGE-depleted magma. Modeling of parental magma in the western intrusion, segment II-W and segment II-E suggests that they were formed by the same initial picritic basalt (100 ppm Cu, 1 ppb Ir and 10 ppb Pd) with different prior sulfide segregations (0.0075%, 0.0085% and 0.011%). The three parts of Jinchuan sulfides show that the Pt/Pd and (Pt + Pd)/(Ir + Ru + Rh) ratios decrease from section III-5 toward both sides in the western intrusion and decrease from section II-14 toward all sides, whereas no regular spatial variations occur in segment II-E, showing the different fractionation processes of sulfide melt. The massive sulfides in the western intrusion and segment II-E experienced a ~20% to 40% and ~40% to 60% fractionation of sulfide melt, respectively. We propose that the Jinchuan deposit was generated in a metallogenic system of multiple magma conduits, where sulfides entrained in parental magma experienced different amounts of prior removal.

**Keywords:** PGE; magmatic sulfides; magma conduit system; Jinchuan deposit; NW China

## 1. Introduction

Magmatic nickel-copper sulfide deposits are generated by the accumulation of an immiscible sulfide liquid which is segregated from a host silicate melt [1–3]. Most magmatic Ni-Cu-(PGE) sulfide deposits were formed in magma conduit systems, such as the Voisey's Bay deposit in Canada [4–6], the Eagle deposit in USA [7,8], and the Huangshandong deposit in western China [9–11]. Because of the preferential partitioning of the chalcophile elements (e.g., PGE, Ni and Cu) into segregated sulfide liquid, the metal tenors of the sulfides are upgraded by the equilibration of successive batches of silicate magma passing through systems of dynamic magma conduits [12–14]. Therefore, the metal tenors in the bulk sulfide ores formed in the magma conduit system were controlled by the magma/sulfide ratio, which reflects the characteristics of magma flow [15–17]. The following fractional crystallization of sulfide melt then leads to the fractionation of the Ir and Pd subgroups of platinum-group elements (IPGE and PPGE) [18,19]. Thus, chalcophile elements are commonly used to study sulfide segregation and mineral fractionation [20–22], which are important to understand sulfide concentration processes and the evolution of mafic-ultramafic rocks in magma conduit systems [15,23,24].

The Jinchuan deposit, one of the three largest Ni-Cu-(PGE) deposits in the world, contains nearly 550 million tons of ore, with 1.1 wt.% Ni and 0.7 wt.% Cu [13,25]. At the Jinchuan deposit, the vast accumulation of sulfide occurred in a small ultramafic body without sulfur-rich country rocks. Researchers increasingly believe that the Jinchuan intrusion resulted from multiple injections of sulfide-charged magma in a system of magmatic conduits [3,26–28]. Chai and Naldrett [29,30] suggested that the Jinchuan deposit was formed by multiple pulses of sulfide-charged magma with different R-factors (ratio of silicate magma to sulfide melt), and orebodies were formed at depth by in situ magma differentiation. Tang and Li [31], De Waal et al. [32] and Song et al. [13,33] proposed that the Jinchuan deposit was formed by injections of sulfide-free and sulfide-bearing olivine mush from a deep-seated staging chamber. Other previous studies have improved the understanding of the genesis of the Jinchuan deposit, especially concerning sulfur saturation and olivine crystallization in the staging magma chamber [32,34]. Recent studies have suggested that the Jinchuan intrusion was originally two individual intrusions, referred to as the western and eastern intrusions [35,36]. Moreover, the occurrence of the intrusion [31] and internal distribution of rocks types [37] may indicate that the Jinchuan intrusion was initially intruded into two sub-vertical structures [28]. However, the differences in magma evolution between these two individual sub-vertical intrusions are still not well understood. Furthermore, for the two largest orebodies in the eastern intrusion, sulfides are enriched in different directions [37] and the average contents of PGEs show an order of magnitude difference [29]. Whether they were formed by the same magma conduit or not is still not clear.

In this study, we collect more extensive and deeper samples from the Jinchuan deposit to compare the PGE tenors in each segment and show a more complete spatial variation of PGE tenors. Our data, together with published data, are used to discuss the sulfide segregation and fractionation processes, and the lateral variations of PGE tenors in different segments of the Jinchuan deposit. Massive sulfide ores are compared with those of the Voisey's Bay deposit in order to understand the fractionation of sulfide melt in the later magmatic metallogenic period. This study demonstrates a new metallogenic model related to three individual magma conduits for the Jinchuan deposit, which provides new insights into the relationship between dyke-like mafic-ultramafic intrusions and metallogenic magma conduit systems.

## 2. Geological Setting

The Jinchuan intrusion is located in the southwest of the Alxa block, in the southwest of the North China Craton (Figure 1a). The Precambrian basement of the Alxa block consists predominantly of 1.9 Ga granitic gneiss and 1.7–2.7 Ga amphibolites [38–40], which are overlain by Cambrian to middle Ordovician sedimentary sequences. The Longshoushan terrane (Figure 1b), which lies in the southwestern margin of the Alxa block, is a southeastern strike uplift [41]. The Longshoushan terrane is bound by two regional thrust faults on its northern ($F_1$) and southern sides ($F_2$) (Figure 1b). In the

Longshoushan terrane, the Paleoproterozoic and Mesoproterozoic metamorphic units (migmatites, gneisses and marbles), are unconformably overlain by the Neoproterozoic and Paleozoic conglomerate, sandstone and limestone. Faults in the Longshoushan terrane are mainly NW- and NE-trending structures, cutting the metamorphic formation [35]. Paleozoic granitoids widely occur in the Longshoushan terrane (Figure 1b) and are associated with the collision of the North Qilian block and the southern Alxa Block roughly 445 Ma [42]. Available zircon U-Pb isotope data suggest that the Jinchuan ultramafic intrusion took place roughly 830 Ma [43,44]. The Xijing mafic intrusion and dolerite dykes occurred roughly 420 Ma [45,46]. Until now, economic sulfide mineralization has only been reported in the Jinchuan intrusion (Figure 1b).

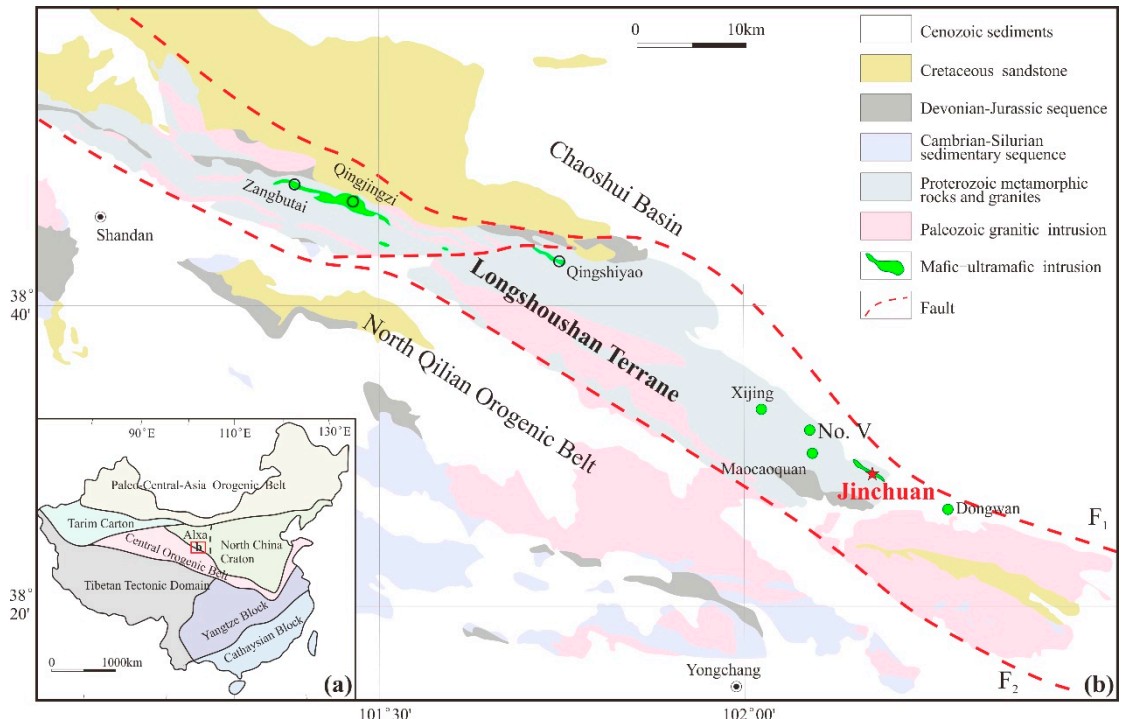

**Figure 1.** (**a**) The location of the Jinchuan Ni-Cu deposit in China and (**b**) a simplified geological map of the Longshoushan terrane. Both subfigures are based on [13].

The Jinchuan ultramafic intrusion intruded the Proterozoic migmatites marbles and gneisses. It is about 6500 m long, 20–527 m wide on the surface (Figure 2a) and has a northwestern downward extension of more than 1000 m from the ground surface (Figure 2b), with dip angle ranging from $50°$ to $80°$ (Figure 2c). Three northeast strike-slip faults (i.e., $F_8$, $F_{16-1}$ and $F_{23}$) divide the Jinchuan intrusion into segments I to IV, with segments III and I being in the west and II and IV in the east (Figure 2a). The western intrusion (to the west of $F_{16-1}$) was emplaced into Proterozoic gneiss and marble, while the eastern intrusion was intruded into Proterozoic marble and migmatites (Figure 2a).

The western intrusion (segments III and I) is narrower (Figure 2a) and is comprised of two rock subunits. The upper part consists of fine-grained dunite, lherzolite and minor pyroxenite from bottom to top, while the lower part consists of coarse-grained dunite and lherzolite. The sulfide mineralization is mainly concentrated at the base of the western intrusion and is subdivided into two orebodies numbered III-1 and I-24 by $F_8$ (Figure 2b). Both of the orebodies consist of net-textured sulfide at the bottom and are overlain by disseminated sulfide.

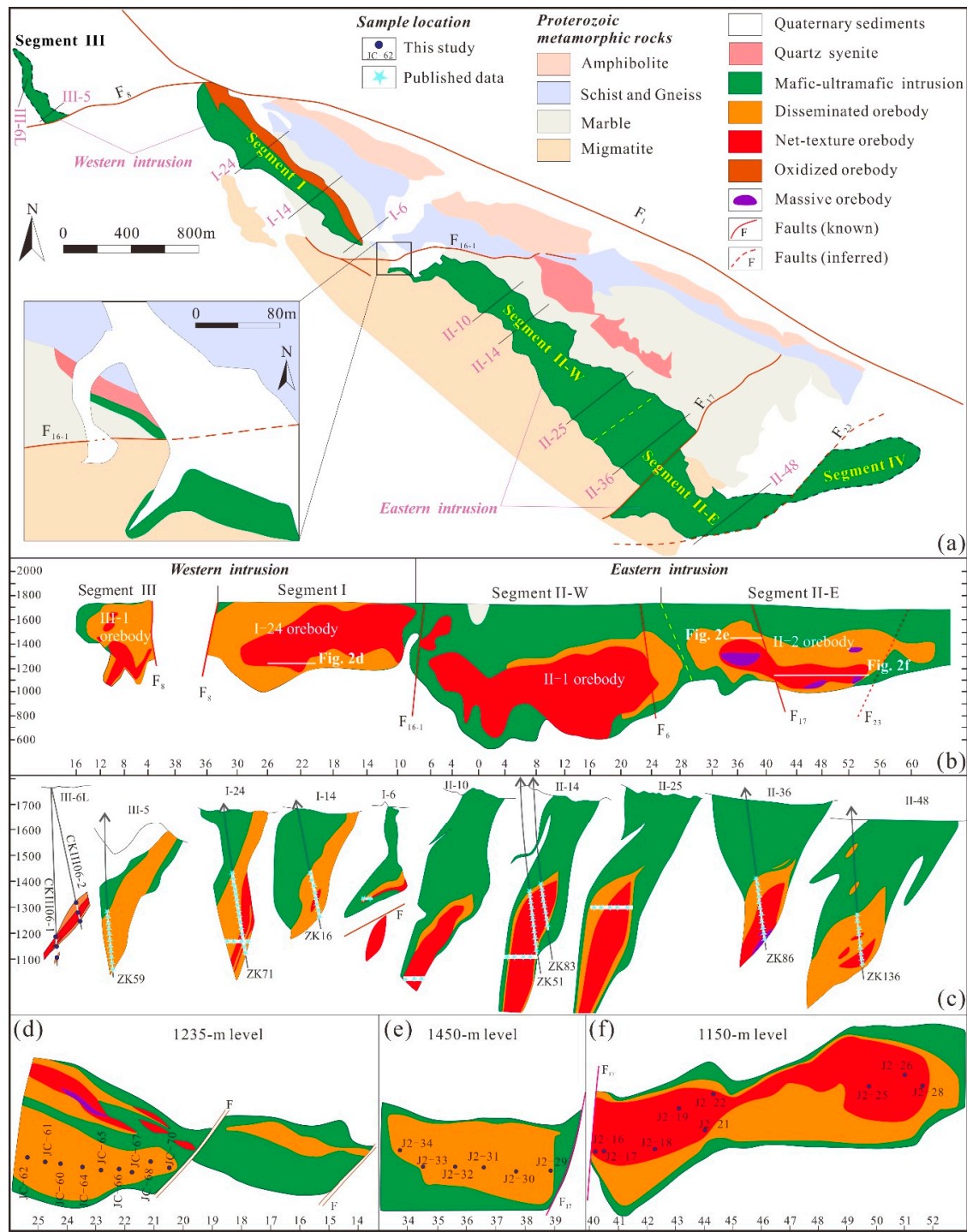

**Figure 2.** Geological map of the Jinchuan deposit (**a**), a projected long section (**b**), selected cross sections with sample locations (**c**) and selected geological plans with sample locations (**d**–**f**).

The eastern intrusion (segments II-W and II-E) is much wider than the western intrusion (Figure 2a). The ultramafic intrusion in the western part of segment II is narrow and tabular with a symmetrical section, but is wider in the eastern part and plunges deeper in the west than in the east. The eastern part of segment II is shallower but wider than the western part, with a V-shaped cross section (Figure 2c). Segment II is further divided into segment II-W and segment II-E, bound by the sulfide-barren lherzolite between orebodies II-1 and II-2 (Figure 2b). Segment II-W is characterized by a concentric distribution of rock types, where the net-textured and disseminated olivine-sulfide

cumulate is surrounded by the dominant coarse-grained lherzolite. In segment II-E, the medium-to coarse-grained olivine-sulfide cumulate and sulfide lherzolite are surrounded by sulfide-barren lherzolite. In segment II-W, sulfides are concentrated toward the center, while in segment II-E, sulfides are concentrated towards the northwestern contact (Figure 2c). Several massive orebodies have been discovered in segment II-E, yet none have been found in segment II-W (Figure 2b).

## 3. Petrology and Sulfide Mineralization of the Jinchuan Deposit

The Jinchuan intrusion is predominantly comprised of dunite, lherzolite, and minor olivine pyroxenite and plagioclase lherzolite. The different types of rocks show a gradational contact relationship [30,31]. Lherzolite is the most abundant rock type in the Jinchuan intrusion and is composed of about 40% to 85% olivine and 10% to 50% pyroxenes (orthopyroxene > clinopyroxene). Dunite, which is composed of more than 90% olivine and less than 10% pyroxene, is commonly observed in the sulfide mineralization zones [32]. Olivine pyroxenite contains about 10% to 40% olivine and 60% to 90% pyroxenes (clinopyroxene > orthopyroxene). Minor amounts of plagioclase lherzolite appear on the surface of the western intrusion, however, it is more widespread in the eastern intrusion and shrinks westward [2].

The primary types of sulfide mineralization in the Jinchuan intrusion are weakly disseminated sulfide, disseminated sulfide (Figure 3a), net-textured sulfide (Figure 3b), massive sulfide (Figure 3c) and Cu-rich sulfide, with a magmatic sulfide mineral assemblage of chalcopyrite, pyrrhotite and pentlandite (Figure 3e,f). Widespread disseminated sulfides and net-textured sulfides are the major ore types of the Jinchuan deposit and are characterized by spot- or net-like sulfides that are interstitial to the cumulus olivine. Massive sulfide ores have relatively higher contents of about 4–9 wt.% Ni and >15% wt.% S and are randomly distributed in segment II-E and the eastern part of segment I (Figure 2b). The Cu-rich sulfide ores, which are recognized by a relatively high amount of chalcopyrite (chalcopyrite: Pyrrhotite: Pentlandite = 3:2.5:1, modal %; [36]), usually occur at the bottom of the eastern segment I.

## 4. Sampling and Analytical Methods

In order to study the lateral variation of trace elements, especially chalcophile elements, in the western intrusion, and segments II-W and II-E of the Jinchuan deposit, we collected samples from strike drift in the deeper parts of segment I and segment II-E, including nine samples (JC-60 to JC-69) from a 1235 m depth in orebody I-24, six samples (J2-29 to J2-34) from a 1450 m depth in western orebody II-2 and thirteen samples (J2-16 to J2-28) from a 1050 m depth in eastern orebody II-2. Segment III is represented by twelve samples (named beginning with JL-) from a deep part of middle segment III (CKIII06-1 and CKIII06-2). All of our samples correspond to about 450–650 m below the surface and are disseminated and net-textured sulfide (except three massive sulfide samples and one Cu-rich sulfide sample), which formed in the same period of magmatic activity. In addition, three samples (named beginning with JLL-) from 1554 m in orebody III-1, seven samples (JJK-01 to JJK-05, JC-14 to JC-18) from the western mining stope of segment II-W and six samples (J2-3 to J2-13) from the eastern mining stope of segment II-E, together with the samples above, were collected to study trace element variation between the different segments. All sulfide mineralization types of the deposit and major rock types of the host intrusion are included in this study.

Analyses of whole-rock major and trace element compositions were performed at the ALS Chemex Co Ltd, Guangzhou China. The rock samples were powdered to 200 mesh under freezing temperature conditions. Major element compositions were determined by X-ray fluorescence (XRF) after dissolution by lithium borate and lithium nitrate. Reference standards GBW07105 and SARM-4 were used to monitor the major element analyses. Trace element compositions were detected using the lithium borate dissolution method, followed by inductively coupled plasma-mass spectrometry (ICP-MS). Standards OREAS-120 and OREAS-100a were used as reference materials to monitor the trace element

analyses. The detection limit and relative standard deviations (RSD, %) for the major and trace elements are listed in Table S1.

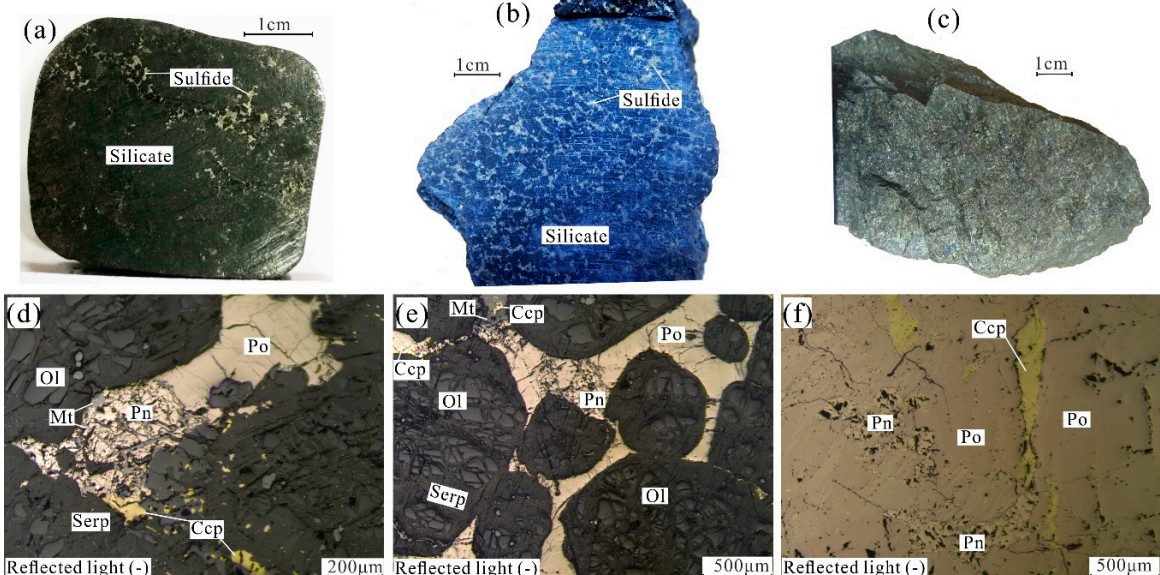

**Figure 3.** Photographs showing the characteristics of the Jinchuan ore types and photomicrographs of sulfide minerals. (**a**) Spot-like sulfide interstitial to cumulus silicate minerals formed in the disseminated sulfide ore. (**b**) Sulfides and intergranular silicates interconnect to form net-textured sulfide ore. (**c**) Massive sulfide ore. (**d**–**f**) Sulfide minerals in disseminated, net-textured and massive sulfide ores. Mineral abbreviations: Ol, olivine; Serp, serpentine; Mt, magnetite; Ccp, chalcopyrite; Po, pyrrhotite; Pn, pentlandite.

The concentrations of Cu and Ni in whole rock samples were determined using atomic absorption spectroscopy (AAS) and the whole-rock S contents were detected using the gravimetric method and IR-absorption spectrometry at the Test Center of Nonferrous Metals Geology and Mining Co, Ltd, Guilin, China. Platinum-group elements (except Os) in whole rock were determined by the nickel sulfide fire assay method using inductively coupled plasma-mass spectrometry, and Os was determined by Carius tube digestion isotope dilution, followed by ICP-MS analysis at the same laboratory. The detailed analytical procedures, blank concentrations and detection limits were previously given by He et al. [47] and Qi et al. [48], which we also followed. The detected limits and RSDs of Cu, Ni, S and PGE are listed in Table 1.

Principal component analysis (PCA) and partial least squares-discriminant analysis (PLS-DA) are multivariate statistical exploratory techniques to interpret and discriminate high-dimensional datasets. In this study, the whole rock geochemistry analyses (69 samples from the western intrusion and 32 samples from the eastern intrusion) from the Jinchuan deposit were investigated by the two methods in the R programming language. Due to the necessary requirement for complete datasets, geochemical data below the detection limit were assigned as the evaluated value given from the laboratory analysis. A detailed description of the PCA and PLS-DA methods we used was previously given by Aitchison [49], Koch [50] and Makvandi et al. [51,52].

## 5. Results

### 5.1. Whole Rock Major and Trace Elements

The concentrations of major elements in the samples from the Jinchuan ultramafic intrusion are given in Table S1. All samples have large losses on ignition (LOI), which is likely associated with hydrothermal alteration and/or sulfide mineralization. Our data, together with previously published data, are used to illustrate the variation of $Mg^{\#}$ ($100MgO/(MgO + FeO^T)$, molar) and loss on ignition

in sulfide-mineralized samples from different segments of the intrusion (Figure 4). The negative correlation between S wt.% versus $Mg^{\#}$ and LOI in sulfide-mineralized samples illustrates that sulfide mineralization rather than hydrothermal alteration is the main contributor to a large LOI. Notably, four segments in the Jinchuan deposit have the same range and trend of variation. This observation indicates that sulfide-mineralized samples (>1 wt.% S) in the four segments underwent similar and limited hydrothermal alteration, which is further evidenced by the original sulfide assemblages (Figure 3). Most of the $Mg^{\#}$ values in the western intrusion are larger than the eastern intrusion (Figure 4a).

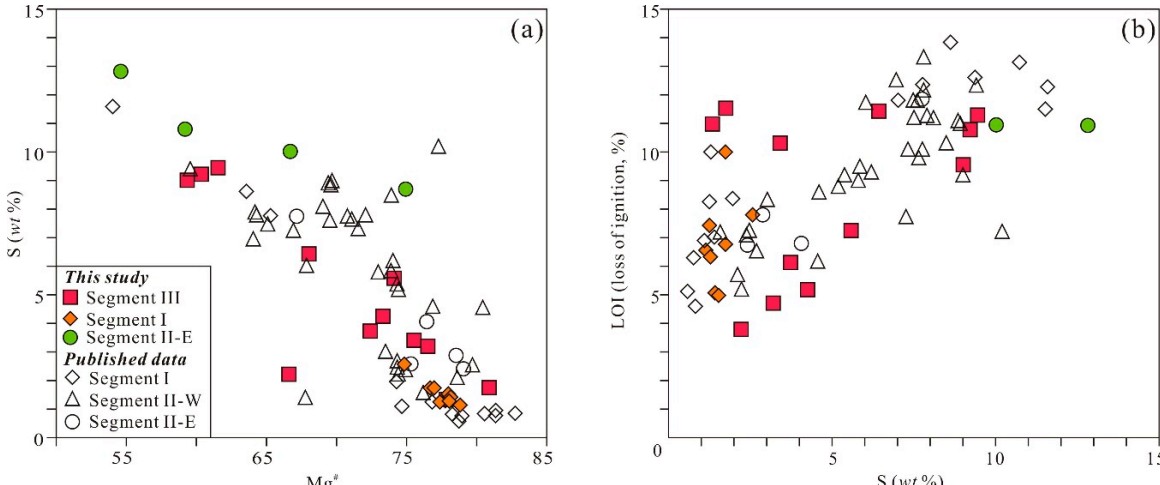

**Figure 4.** Plotting of sulfide mineralization samples $MgO^{\#}$ ($100MgO/(MgO + FeO^T)$, molar) and loss on ignition (LOI) versus S wt.% in whole rock at Jinchuan. Data from this study and [13,17,32,33,35].

The analysis results for trace elements are also given in Table S1. All samples from the Jinchuan intrusion show a similar trace element distribution pattern, characterized by the enrichment of large-ion lithophile elements (LILE) and light rare-earth elements (LREE) (Figure 5). Notably, there are some differences in the concentrations of trace elements (such as high-field-strength elements (HFSE), LILE and rare-earth elements (REE)) between the western and eastern intrusions. In the western intrusion (segments III and I), the concentrations of trace elements of the sulfide-barren (<1 wt.% S) samples have a narrow range within that of the sulfide mineralized (>1 wt.% S) samples (Figure 5a–d). However, samples from the eastern intrusion (segments II-W and II-E) show that the sulfide-barren samples contain higher concentrations of trace elements than the sulfide-mineralized samples (Figure 5e,f). Most samples from the western intrusion have negative Eu and positive Th anomalies, whereas no obvious Eu and Th anomalies occur in the eastern intrusion (Figure 5 and Table S1).

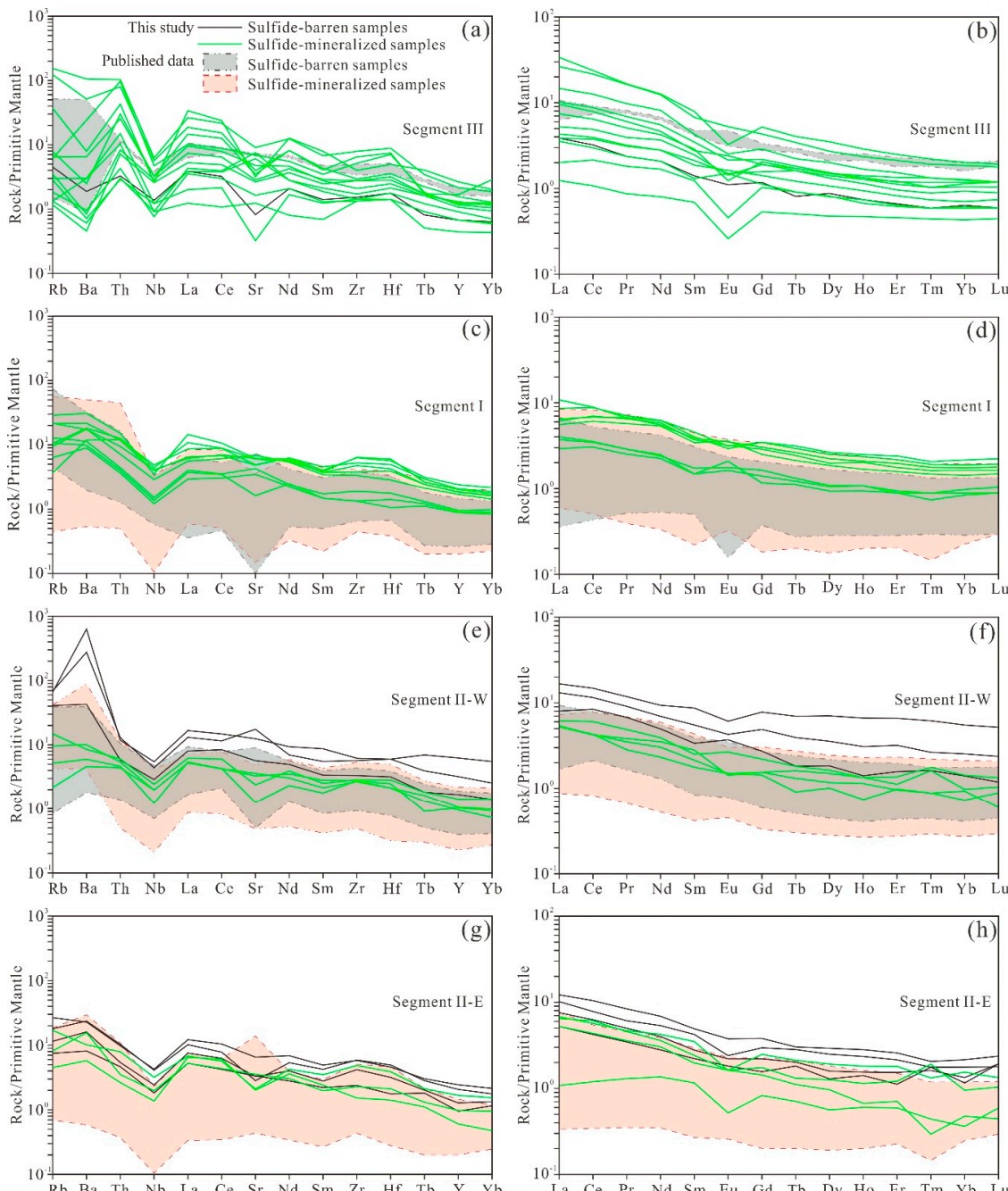

**Figure 5.** Primitive mantle-normalized trace element spider diagrams and rare-earth element (REE) patterns of sulfide-barren and sulfide-mineralized samples from segment III (**a**,**b**), I (**c**,**d**), II-W (**e**,**f**), II-E (**g**,**h**) at Jinchuan. Data from this study and [30,33,35], data of primitive mantle are taken from [53].

### 5.2. Nickel, Copper, and Platinum-Group Elements

The analysis results of the Ni, Cu, PGEs and sulfur in samples from the Jinchuan intrusion are listed in Table 1. All our samples in the Jinchuan intrusion show similar ΣPPGE/ΣIPGE ((Pt + Pd)/(Os + Ir + Ru + Rh)) ratios (mean 11.4), Cu/Pd ratios (mean 172,206) and Pd/Ir ratios, which mostly range from 1.25 to 26.55 (except JL-6 and JL-13). These ratios are larger than the primitive mantle (ΣPPGE/ΣIPGE = 0.8, Cu/Pd = 7690, Pd/Ir = 1.21, [53]). The Cu/Pd ratios of disseminated and net-textured sulfide in segment II-E (mean 165726) are notably higher than those in the western intrusion (mean 71,983) and segment II-W (mean 50,800, [13]). The Ni/Cu ratios of disseminated and net-textured samples are in the range of 0.22–2.63 in segment III, 0.79–6.96 in segment I and 0.85–9.14

in segment II-E. The most impressive correlations are the positive linear covariations of the chalcophile elements with S wt.% in whole rock (Figure 6), which means that the chalcophile elements mostly occur in the sulfide. The concentrations of chalcophile elements in the disseminated sulfide from the western intrusion are slightly higher than in the eastern intrusion, particularly in segment II-E (Figure 6). For the net-textured samples, the concentration of Pd from segment II-E is much lower than both segments I and II-W (Figure 6d).

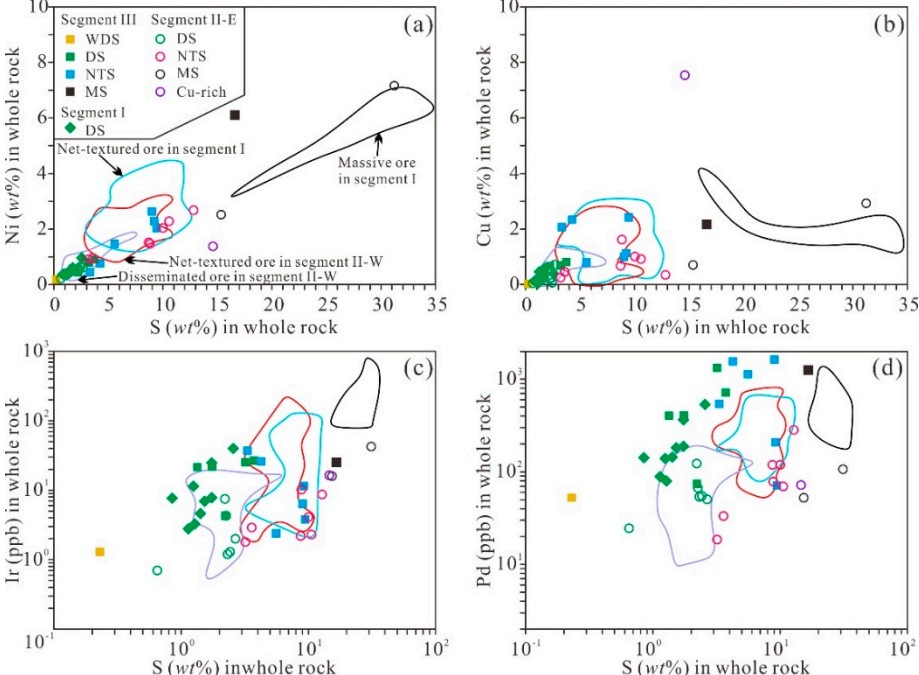

**Figure 6.** Variations of S versus Ni, Cu, Ir and Pd for the samples from the Jinchuan intrusion. Data represented by outlines were taken from [13,16,17,29,33,36,54].

In order to examine the variation of chalcophile elements present in the sulfide minerals in the Jinchuan intrusion, it is necessary to recalculate the concentrations of chalcophile elements in 100% sulfide, commonly referred to as metal tenors in the literature. In this study, we used the Barnes and Lightfoot equation [12] to calculate the metal tenors in the Jinchuan deposit, which is based on the assumption that the magmatic sulfide assemblage is composed of pyrrhotite, pentlandite and chalcopyrite [55]. A few samples (those with less than 0.5 wt.% S) were excluded during the calculation.

After normalization to 100% sulfide, the Jinchuan sulfides showed a clear positive correlation between Ir and Ru (Figure 7a) and a weak positive correlation between Pd and Pt (Figure 7b), but no correlation of Ir or Pd with Cu (Figure 7c,d). The disseminated sulfides are characterized by a narrow range of PPGE/IPGE and weak positive correlation between them (Figure 7e,f). The net-textured sulfides show a clearly negative correlation between IPGE and PPGE in segment II-W, but a positive correlation in segment II-E (Figure 7e,f). Comparing with the disseminated and net-textured sulfide, the massive sulfides were typically depleted in PPGE, while the Cu-rich sulfides were typically depleted in IPGE (Figure 7). The primitive mantle-normalized patterns of the Ni-PGE-Cu tenors in four segments of the Jinchuan deposit are illustrated in Figure 8. The disseminated sulfides of the western intrusion are characterized by being the least fractionated between PPGE and IPGE, which contain higher PGE tenors than net-textured sulfides (Figure 8a,b). However, the disseminated and net-textured sulfides of the eastern intrusion show similar patterns and PGE tenors (Figure 8c,d). All massive sulfides (except JL-27) showed strongly negative Pt anomalies, while the Cu-rich sulfides showed positive anomalies (Figure 8e). The PGE tenors of not only the disseminated sulfide, but also the massive sulfide in the western intrusion, are higher than those in the eastern intrusion (Figures 7 and 8).

**Table 1.** The concentrations of chalcophile elements of samples from the Jinchuan deposit.

| Segment | Sample | Ore Type | Os (ppb) | Ir (ppb) | Ru (ppb) | Rh (ppb) | Pt (ppb) | Pd (ppb) | Cu (wt.%) | Ni (wt.%) | S (wt.%) | Pd/Ir | ΣPPGE/ΣIPGE | Cu/Pd/1000 |
|---|---|---|---|---|---|---|---|---|---|---|---|---|---|---|
| | | Detected limit | 0.007 | 0.013 | 0.02 | 0.001 | 0.026 | 0.06 | 0.0005 | 0.0003 | 0.0001 | | | |
| Relative standard deviation (RSD, %) | | | 4.2 | 3.5 | 4.1 | 3.9 | 2.8 | 3.1 | 0.61 | 0.56 | 0.72 | | | |
| Segment III | JL-6 | NTS | 3.9 | 2.4 | 4.5 | 2.1 | 19.3 | 566.6 | 0.80 | 1.47 | 5.58 | 236.1 | 45.4 | 14.1 |
| | JL-13 | NTS | 15.2 | 6.4 | 11.8 | 6.4 | 76.0 | 819.4 | 1.00 | 2.63 | 9.01 | 128.0 | 22.5 | 12.2 |
| | JL-18 | DS | 44.0 | 26.8 | 32.2 | 11.2 | 255.2 | 360.5 | 0.81 | 0.91 | 3.73 | 13.4 | 5.4 | 22.5 |
| | JL-23 | NTS | 85.1 | 37.2 | 50.8 | 15.3 | 82.2 | 271.2 | 2.07 | 0.45 | 3.32 | 7.3 | 1.9 | 76.3 |
| | JL-27 | MS | 33.6 | 25.2 | 18.4 | 15.6 | 602.1 | 631.3 | 2.17 | 6.11 | 16.61 | 25.0 | 13.3 | 34.4 |
| | JL-38 | DS | 37.4 | 25.4 | 19.3 | 16.3 | 623.6 | 664.4 | 0.69 | 0.80 | 3.20 | 26.2 | 13.1 | 10.4 |
| | JL-50 | NTS | 45.6 | 26.1 | 28.0 | 15.4 | 552.4 | 780.0 | 2.34 | 0.76 | 4.25 | 29.9 | 11.6 | 30.0 |
| | JL-55 | NTS | 6.5 | 3.8 | 5.4 | 1.9 | 36.1 | 35.8 | 2.42 | 2.03 | 9.45 | 9.4 | 4.1 | 676.0 |
| | JL-60 | NTS | 10.3 | 11.5 | 13.0 | 4.6 | 134.8 | 104.6 | 1.13 | 2.28 | 9.22 | 9.1 | 6.1 | 108.0 |
| | JL-66 | DS | 4.7 | 4.3 | 5.0 | 1.7 | 34.3 | 37.3 | 0.67 | 0.44 | 2.22 | 8.7 | 4.6 | 179.6 |
| | JLL-1 | DS | 24.7 | 21.5 | 25.1 | 8.9 | 247.1 | 202.9 | 0.17 | 0.44 | 1.34 | 9.4 | 5.6 | 8.4 |
| | JLL-2 | DS | 25.5 | 21.9 | 25.5 | 9.0 | 249.3 | 203.7 | 0.35 | 0.57 | 1.75 | 9.3 | 5.5 | 17.2 |
| | JLL-3 | WDS | 3.0 | 1.3 | 1.8 | 0.5 | 21.0 | 26.4 | 0.02 | 0.19 | 0.23 | 20.3 | 7.2 | 9.1 |
| Segment I | JC-60 | DS | 6.9 | 7.7 | 6.7 | 2.9 | 52.9 | 71.2 | 0.14 | 0.37 | 0.85 | 9.2 | 5.1 | 19.7 |
| | JC-61 | DS | 30.8 | 40.1 | 32.4 | 13.5 | 119.0 | 268.0 | 0.40 | 0.96 | 2.57 | 6.7 | 3.3 | 14.9 |
| | JC-62 | DS | 18.6 | 24.9 | 21.0 | 9.4 | 362.0 | 183.0 | 0.58 | 0.60 | 1.74 | 7.4 | 7.4 | 31.7 |
| | JC-64 | DS | 9.8 | 11.5 | 9.6 | 4.1 | 241.0 | 70.0 | 0.46 | 0.37 | 1.25 | 6.1 | 8.9 | 65.7 |
| | JC-65 | DS | 6.6 | 6.9 | 6.3 | 3.7 | 266.0 | 91.1 | 0.18 | 0.46 | 1.53 | 13.2 | 15.2 | 19.8 |
| | JC-66 | DS | 6.9 | 7.8 | 6.7 | 3.0 | 157.7 | 94.5 | 0.39 | 0.52 | 1.74 | 12.1 | 10.3 | 41.3 |
| | JC-67 | DS | 4.3 | 4.6 | 4.2 | 2.5 | 400.0 | 72.3 | 0.29 | 0.41 | 1.42 | 15.7 | 30.3 | 40.1 |
| | JC-68 | DS | 3.0 | 3.3 | 2.8 | 1.1 | 53.0 | 39.8 | 0.16 | 0.40 | 1.28 | 12.1 | 9.1 | 40.2 |
| | JC-69 | DS | 2.9 | 2.8 | 2.8 | 1.8 | 57.2 | 44.6 | 0.05 | 0.37 | 1.14 | 15.9 | 9.9 | 11.2 |
| Segment II-E | J2-11 | Cu-rich | 12.7 | 16.4 | 16.7 | 28.8 | 8416.0 | 36.1 | 7.54 | 1.38 | 14.60 | 2.2 | 113.3 | 2088.6 |
| | J2-16 | MS | 83.9 | 42.6 | 129.3 | 34.3 | 63.8 | 53.5 | 2.93 | 7.16 | 31.23 | 1.3 | 0.4 | 547.7 |
| | J2-17 | MS | 33.4 | 16.0 | 41.2 | 12.1 | 11.1 | 26.3 | 0.71 | 2.52 | 15.35 | 1.6 | 0.4 | 270.0 |
| | J2-18 | NTS | 3.4 | 2.3 | 4.5 | 3.9 | 17.2 | 34.9 | 0.92 | 2.28 | 10.58 | 15.2 | 3.7 | 263.6 |
| | J2-20 | NTS | 18.3 | 8.7 | 23.0 | 7.5 | 102.7 | 142.3 | 0.35 | 2.68 | 12.82 | 16.4 | 4.3 | 24.6 |
| | J2-21 | NTS | 4.3 | 2.2 | 4.7 | 2.3 | 11.4 | 59.7 | 0.68 | 1.52 | 8.71 | 27.1 | 5.3 | 113.9 |
| | J2-22 | NTS | 21.2 | 10.2 | 30.0 | 9.4 | 191.1 | 39.3 | 1.62 | 1.47 | 8.83 | 3.8 | 3.3 | 412.2 |
| | J2-25 | NTS | 5.9 | 4.1 | 9.7 | 5.3 | 3.9 | 59.7 | 1.01 | 2.04 | 10.02 | 14.6 | 2.5 | 169.2 |
| | J2-26 | NTS | 3.2 | 1.8 | 3.3 | 1.6 | 30.4 | 9.3 | 0.25 | 0.91 | 3.20 | 5.2 | 4.0 | 268.8 |
| | J2-28 | NTS | 3.9 | 2.9 | 4.0 | 3.3 | 32.8 | 16.7 | 0.46 | 0.96 | 3.59 | 5.8 | 3.5 | 275.4 |
| | J2-29 | DS | 2.3 | 2.0 | 3.4 | 2.5 | 9.2 | 25.3 | 0.72 | 0.61 | 2.67 | 12.6 | 3.4 | 284.6 |
| | J2-30 | DS | 1.9 | 1.2 | 2.8 | 1.1 | 20.7 | 27.0 | 0.24 | 0.55 | 2.32 | 22.5 | 6.8 | 88.9 |
| | J2-31 | DS | 2.2 | 1.3 | 3.0 | 1.8 | 256.2 | 27.6 | 0.07 | 0.64 | 2.43 | 21.2 | 34.2 | 25.4 |
| | J2-32 | DS | 4.9 | 4.3 | 6.3 | 2.8 | 25.3 | 33.6 | 0.52 | 0.50 | 2.26 | 7.8 | 3.2 | 154.8 |
| | J2-33 | DS | 9.8 | 7.5 | 11.6 | 3.4 | 124.4 | 61.7 | 0.20 | 0.58 | 2.21 | 8.2 | 5.8 | 32.4 |
| | J2-34 | DS | 1.0 | 0.7 | 1.7 | 1.0 | 14.0 | 12.3 | 0.05 | 0.22 | 0.65 | 17.6 | 6.0 | 40.6 |

Abbreviations: NTS, net-textured sulfide; DS, disseminated sulfide; MS, massive sulfide; WDS, weakly disseminated sulfide; Cu-rich, Cu-rich sulfide.

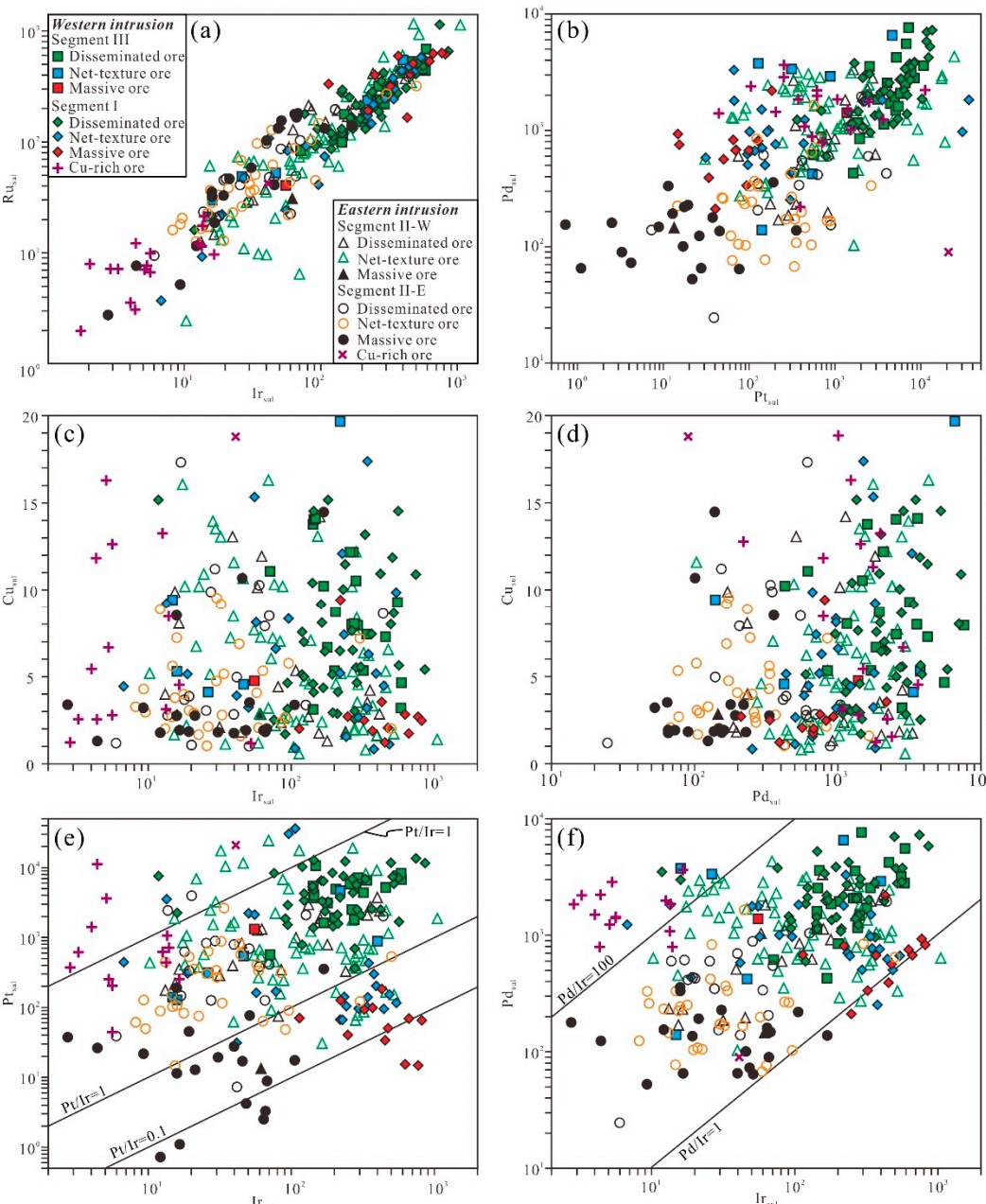

**Figure 7.** Variations of Ru versus Ir (**a**), Pd versus Pt (**b**), Cu versus Ir (**c**), Cu versus Pd (**d**), and Pt and Pd versus Ir (**e**,**f**) normalized to 100% sulfides for the samples from the Jinchuan deposit. Data are from this study and [13,16,17,29,33,36,54].

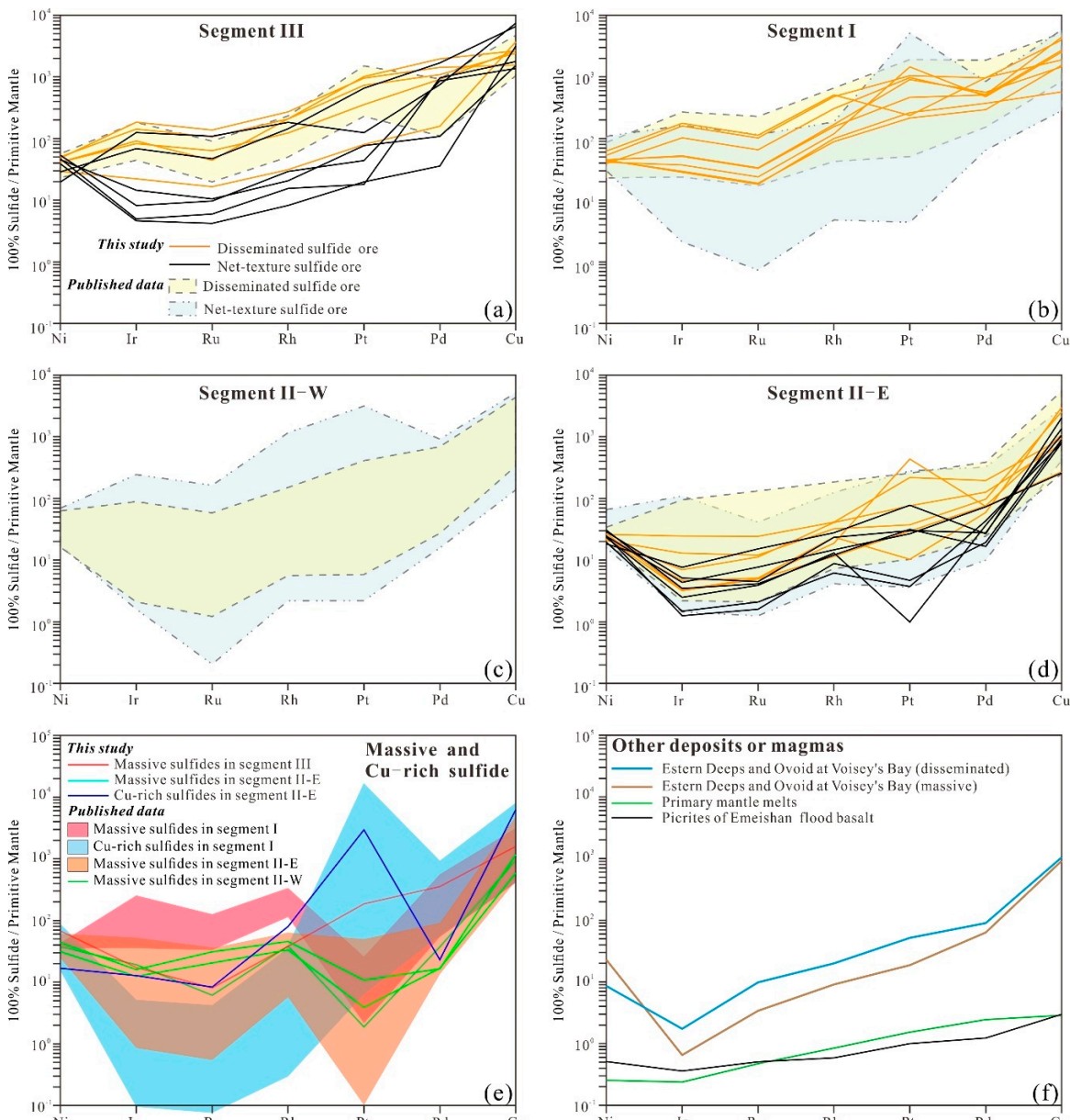

**Figure 8.** Primitive-mantle normalized Ni, PGE and Cu patterns of sulfides in the Jinchuan deposit (**a–e**) and other deposits or magmas (**f**) normalized to 100% sulfide. Data for the Jinchuan deposit are from this study and [13,16,17,29,33,36,54]. The primitive mantle values of metals used in the normalization are from [53]. Data for the Eastern Deeps and Ovoid of Voisey's Bay deposit are from [56,57], those for the primitive mantle melts are from [12], and those for the picrite of Emeishan flood basalt are from [58].

## 5.3. PCA and PLS-DA Results

The results of the PCA and PLS-DA applied to the trace element data are shown in Figure 9. In Figure 9a, principal component 1 (PC1) accounts for 78.4%, including most of element concentrations that are positively related, whereas principal component 2 (PC2), as a measure of Ba, accounts for 8.6%. Spots of the western intrusion generally represent a weakly positive variation in the PC1 versus PC2 plane, which is in contrast with the trending of spots of the eastern intrusion (Figure 9a). In the score scatter plots ($t_1$ versus $t_2$) of PLS-DA (Figure 9b), the significantly different variations in the two intrusions are shown: Samples from the western intrusion are mainly plotted in the negative $t_2$ region, with a positive relationship between $t_1$ and $t_2$, whereas samples from the eastern intrusion are typically

plotted in the positive $t_2$ side, with a negative correlation. Thus, the PCA and PLS-DA results indicate that the trace element concentrations are good discriminators for the two intrusions.

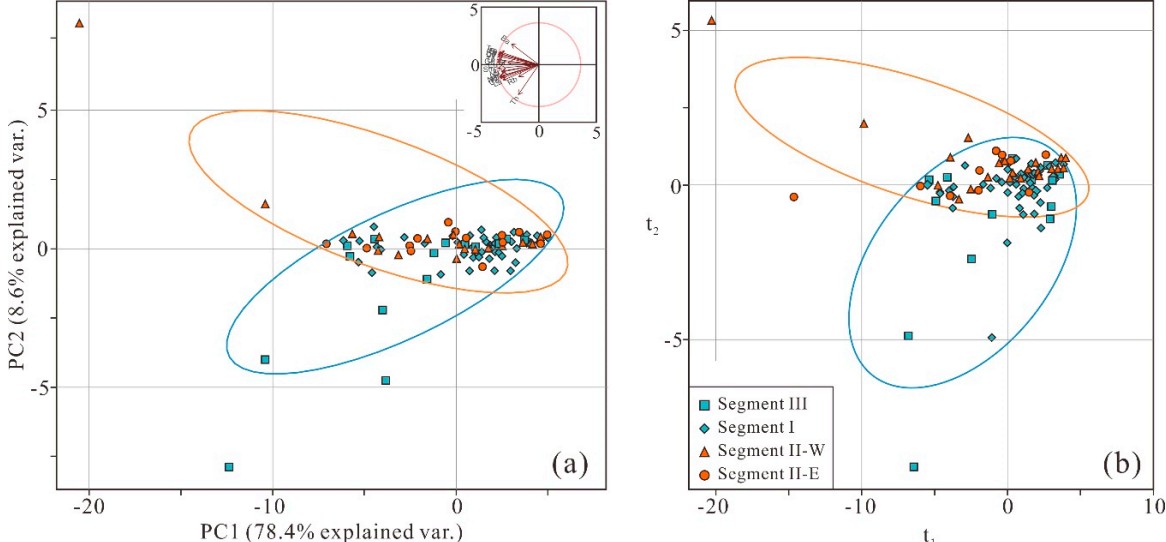

**Figure 9.** Principal component analysis (PCA) and partial least squares-discriminant analysis (PLS-DA) of trace elements at the Jinchuan deposit. (**a**) Plots of PC1 versus PC2 plane (explaining 87.0% of trace element content variability); (**b**) plots of $t_1$ versus $t_2$ (first and second scores), showing the distributions of the western and eastern intrusions.

## 6. Discussion

### 6.1. Prior Sulfide Segregation

The compositions of parental magmas play a critical role in the PGE compositions of the sulfides that segregate from them [12,59,60]. The negative correlation between S wt.% versus Mg[#] and LOI demonstrates that there is no difference in the degree of hydrothermal alteration between the four segments. The disseminated samples of the Jinchuan intrusion show very narrow Pd/Ir ratios (Figure 7f), which are mostly plotted in the range of typical sulfides segregated from picritic magma (Figure 10a). The Ni/Pd versus Cu/Ir diagram shows most samples are in the range of high Mg basalt (Figure 10b). The genetic relationships are also suggested by the Ni/Cu ratios, which are lower than that segregated from komatiitic magmas (>7, [2]), but comparable with basalt magmas such as those from the Nadezhdinsky Formation at the Noril'sk region (~1, [61–63]) and Sudbury in Canada (1.2, [12]). The MgO content of parental magma of the Jinchuan intrusion was estimated to be 11.5–12.6 wt.% [17,30,64]. Thus, we propose that the primitive magma of the Jinchuan intrusion had the composition of high MgO picritic basalt and contained about 100 ppm Cu, 1 ppb Ir and 10 ppb Pd. The initial contents of Cu, Ir and Pb are within the range of data from typical undepleted picritic basalts of Qeqertarssuaq, West Greenland (80–1400 ppm Ni, 0.43–1.35 ppb Ir, 0.23–3.16 ppb Ru, 0.26–0.58 ppb Rh, 5.7–9.4 ppb Pt and 4.2–12.9 ppb Pd; [65]), and are similar to picritic basalts in primitive mantle melt (86 ppm Cu, 498 ppm Ni, 0.76 ppb Ir, 2.36 ppb Ru, 0.76 ppb Rh, 10.86 ppb Pt and 9.54 ppb Pd; [12]).

The Cu/Pd ratios of the Jinchuan intrusion (Table 1) are much higher than primitive mantle (~7700, [53]), suggesting that PGEs were depleted due to prior sulfide segregation [66–68]. Prior sulfide segregation is also suggested by the significantly lower Ir and Pd contents of the sulfide-barren rocks in the Jinchuan intrusion [13] compared to the siliceous high magnesium basalt of the Archaean Yilgarn Craton [69]. The low forsterite contents of olivine [32,37] and a strong linear correspondence of molecular (MgO + FeO)/Al₂O₃ versus SiO₂/Al₂O₃ and SiO₂/TiO₂, with slopes of ~2 [30], suggest that the parental magmas experienced significant fractional crystallization before being intruded at Jinchuan, which may also have triggered the sulfide segregation [2]. The separated sulfide melts

extracted chalcophile elements and then aggregated together at the base of the staging magma chamber under the action of gravity. Thus, the concentration of an element in the residual melt of a fractionating magma can be determined from the Rayleigh fractionation equation [70]:

$$C_L = C_0 \times (1 - F)^{(D - 1)},\qquad(1)$$

where $C_L$ denotes the concentration of the element in the fractionated magma, $C_0$ is the concentration in the initial magma, D is the partition coefficient of the element between the sulfide and the silicate liquid and F represents the fraction of liquid sulfide that is segregated. In this study, $D_{Pt}$, $D_{Pd}$, $D_{Cu}$ and $D_{Ir}$ were defined as 40,000, 40,000, 1000 and 30,000, respectively, which are within the range of the experimental results [67,71,72].

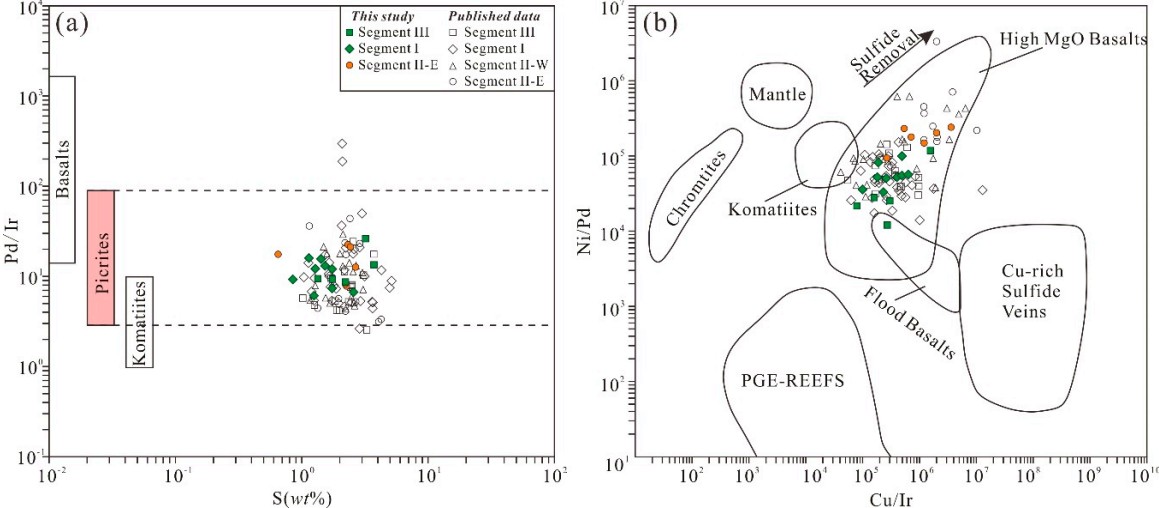

**Figure 10.** Plots of Pd/Ir versus S wt.% (**a**) and Ni/Pd versus Cu/Ir (**b**) of the disseminated ores in the Jinchuan deposit, based on [36,73]. The published data were taken from [13,16,17,29,33,36,54].

As mentioned above, the Jinchuan intrusion consists of two individual intrusions. Not only the spatial distribution and chalcophile element compositions of sulfide mineralization, but also the Eu and Th anomalies and content of trace elements in the western intrusion were significantly different from the eastern intrusion. Our statistical results show that the PCA and PLS-DA (Figure 9) clearly distinguished the two intrusions. Additionally, the La/Sm ratios of the eastern intrusion (2–4.7) show a narrower range than the western intrusion (0.6–6.3). These differences likely imply distinctly different parental magmas and fractionation processes between the western and eastern intrusions. Moreover, the Cu/Pd ratios in segment II-E are obviously higher than those of segment II-W, which suggests a higher amount of prior sulfide removal [66]. Different amounts of prior sulfide removal are also suggested by the negative relationship between Ir and Pd of net-textured sulfide in segment II-W, but a positive relationship in segment II-E. In this study, we estimated the depletion factor of three parts (western intrusion, segment II-W and segment II-E) of the intrusion by comparing the measured Cu/Pd ratios (Figure 11) to simulations for the process of sulfide segregation. Based on Equation (1), we propose that the western intrusion experienced a ~0.0075% segregation of sulfide melt, resulting in the depletion of chalcophile metals in the residual magma, with 0.50 ppb Pd, 0.11 ppb Ir and 79.79 ppm Cu. However, for the eastern intrusion, segment II-W may have experienced a ~0.0085% segregation of sulfide melt, resulting in residual magma containing 0.33 ppb Pd, 0.08 ppb Ir and 79.00 ppm Cu. Segment II-E may have experienced a ~0.011% segregation of sulfide melt, resulting in residual magma with 0.12 ppb Pd, 0.04 ppb Ir and 77.05 ppm Cu. The concentrations of chalcophile elements in the residual magma of segments I and II-W are close to those of dunite from the western intrusion [36] and lherzolite from the eastern intrusion [13], respectively. However, the Cu/Pd ratio of the residual

magma from segment II-E is close to the disseminated and net-textured sulfide. The higher degree of sulfide segregation in the disseminated and net-textured sulfide from segment II-E was probably caused by a different process of magma emplacement from segments I and II-W. Therefore, we suggest that different amounts of prior sulfide segregation resulted in the depletion of PGEs from the parental magmas of these parts.

*6.2. Spatial Variation of R-Factor and PGE Tenors*

PGE tenors in sulfide are determined by the concentration of PGEs in parental magma and the ratio of silicate magma to sulfide melt (R-factor) [59]. Thus, the PGE tenors of the disseminated and net-textured sulfide formed by the PGE-depleted magma at Jinchuan are suitably modeled using the R-factor, which is described by the following equation [59]:

$$Y_i = X_{i0} \times D_i \times (R + 1)/(R + D_i), \tag{2}$$

Here, $Y_i$ and $X_{i0}$ represent the concentrations of element i in the sulfide melt and silicate liquid, respectively. $D_i$ represents the sulfide/silicate liquid partition coefficient of i and R is the mass ratio of silicate melt to that of segregated sulfide melt. Barnes et al. [66] proposed that the PGE tenors of Ni-Cu sulfide deposits and PGE reefs can be modeled using Cu/Pd ratios because of the difference between the partition coefficients for Cu and Pd in liquid sulfide. Based on Equation (2), the relationship between the Cu/Pd ratios and R-factor can be expressed as:

$$Y_{Cu}/Y_{Pd} = X_{Cu0}/X_{Pd0} \times D_{Cu}/D_{Pd} \times (R + D_{Pd})/(R + D_{Cu}) \tag{3}$$

The results of modeling show that three parts of the Jinchuan sulfides can be explained by R-factors ranging from 100 to 50,000, where most samples are plotted in the range of 1000 to 10,000 (Figure 11), after accounting for different depletion factors due to sulfide segregation. Studies on the Bushveld Complex show that when R is in the range of 100 to 2000, the Cu content of the sulfide is typical for most Ni sulfide ores, and the PGE tenors are relatively low (<0.5 ppm for Pt and Pd), however, when R is in the range of 10,000 to 100,000, the Cu content is not much higher than that for a R-factor less than 10,000, but the amount of PGE tenors would be much higher [2]. The PGE concentrations in parental magma of Jinchuan are comparable with Voisey's Bay (0.5 ppb Pd, [57]). However, the R-factor of Jinchuan is larger than Voisey's Bay (100 to 5000, [56,57]), possibly resulting in the higher amount of PGE tenors in the disseminated sulfide (Figure 8f). It is noted that no PGE ores were found at the Jinchuan deposit, even though a few samples have R-factors greater than 10,000. This absence of PGE ores is most likely a result from the lower PGE concentration of parental magma.

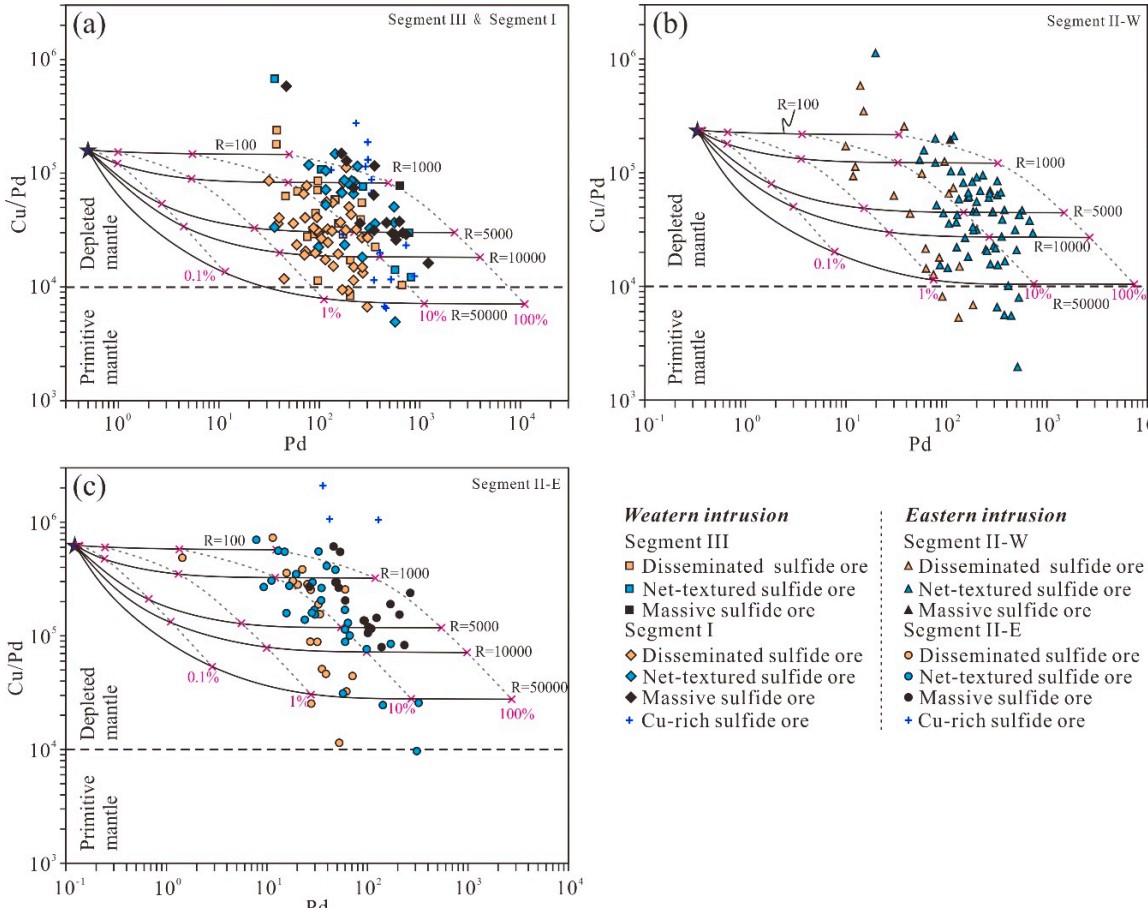

**Figure 11.** The relationship between Cu/Pd concentrations of the Jinchuan magmas and R-factors for the Jinchuan sulfides. Data are from this study and [13,16,17,29,33,36,54].

In order to estimate the spatial variation of PGE tenors in the western intrusion, segments II-W and II-E of the Jinchuan intrusion, the variation of Pt/Pd and the (Pt + Pd)/(Ir + Ru + Rh) ratios of the disseminated and net-textured sulfide samples along the striking trend were illustrated, as seen in Figure 12a. Both of the two ratios decrease from section III-5 towards both sides in the western intrusion and decrease from section II-14, as a center, toward all sides in segment II-W, whereas no regular spatial variations occur in segment II-E (Figure 12). The zonation of Pt/Pd and the (Pt + Pd)/(Ir + Ru + Rh) ratios, which are controlled by the fractionation of sulfide melt [74–76], may reflect the direction of magma flow. The immiscible sulfide segregation from magma and the fractional crystallization of the monosulfide solid solution (MMS), evaluated using a plot of Pd versus Ir [77], displays the similar variation trends to the Pt/Pd and (Pt + Pd)/(Ir + Ru + Rh) ratios (Figure 12e–g). The lack of obvious lateral variation of metal ratios in segment II-E may result from a wider magma conduit or uniform sulfide melt deposition. In summary, the lateral variation of metal ratios and internal R-factors have different variation trends in the western intrusion and segments II-W and II-E of the Jinchuan deposit.

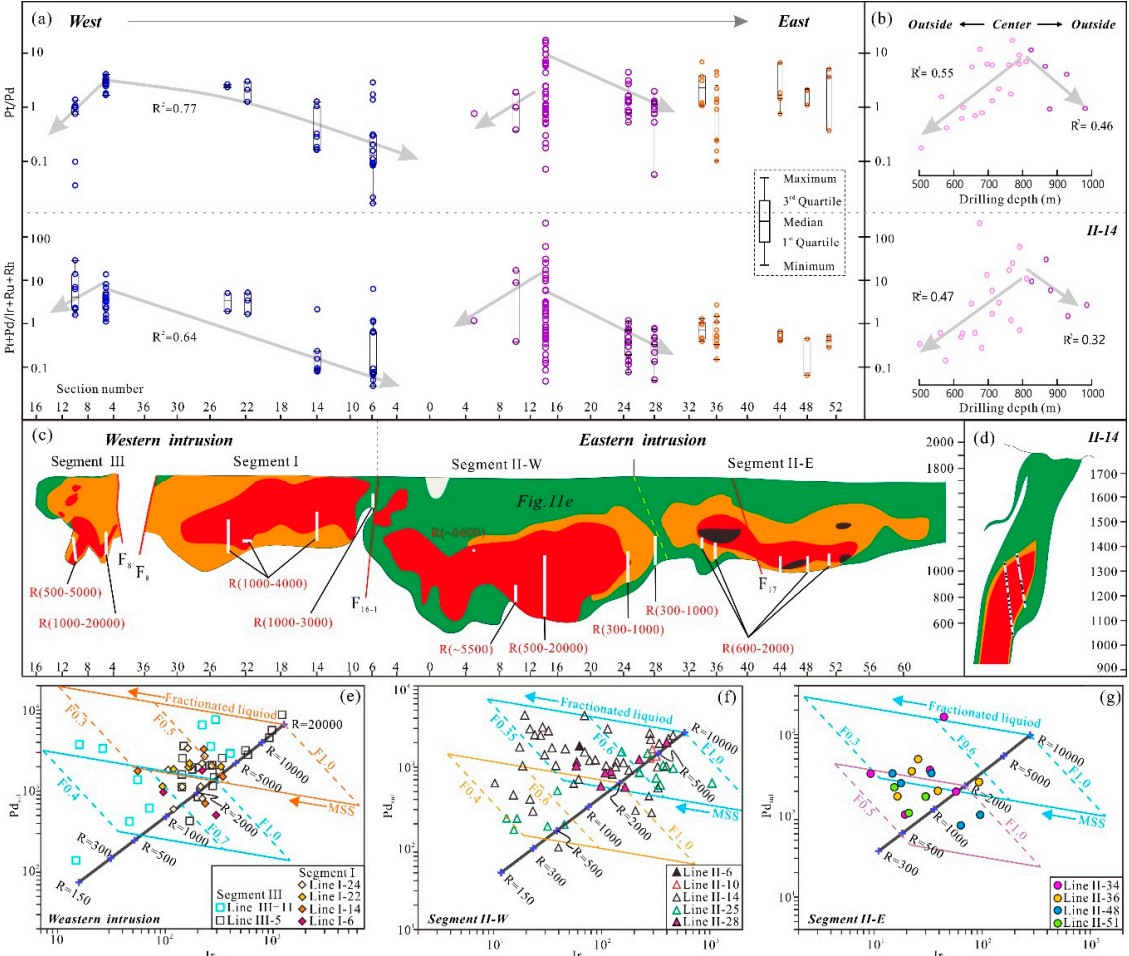

**Figure 12.** Variation of Pt/Pd and the (Pt + Pd)/(Ir + Ru + Rh) ratios along the strike (**a**) and selected drill hole (**b**) and the corresponding sample location in the Jinchuan deposit (**c,d**), followed by modeling of the Pd and Ir tenors of those samples in the western intrusion (**e**), segment II-W (**f**), and segment II-E (**g**). The data are from this study and [13,17,29,36,54]. Modeling parameters: $D_{Ir}^{sulfide/magma}$ = 30,000, $D_{Pd}^{sulfide/magma}$ = 40,000 [72]; $D_{Ir}^{MSS/sulfide}$ = 5, $D_{Pd}^{MSS/sulfide}$ = 0.1 [19]; initial magma, Ir = 0.11 ppb and Pd = 0.50 ppb for the western intrusion, Ir = 0.078 ppb and Pd = 0.33 ppb for segment II-W and Ir = 0.037 ppb and Pd = 0.12 ppb for segment II-E.

### 6.3. Massive Sulfide Fractionation at Jinchuan

Our results show that there is a negative correlation between IPGE and PPGE in segment II-W (Figure 7e,f), and a relatively lower PPGE content of massive sulfide (Figure 8), indicating a significant fractionation of sulfide melts. For the Cu-rich sulfide, the high PPGE tenors and extremely low IPGE tenors (Figures 7 and 8) also suggest a decoupling between IPGE and PPGE. The proportions of cumulus monosulfide solid solution (MSS), and of the liquid from which the MSS has crystallized in any sample, can be determined by plotting the concentration of a compatible element (e.g., Ir, Ru, Rh or Os) against one that is incompatible (such as Cu, Pt or Pd) in the MSS [19,78]. In this study, we used the Rayleigh equation (Equation (1)) to model the fractionation of Rh and Cu between MSS and a sulfide melt, using the MSS/sulfide melt partition coefficients of $D_{Rh}^{MSS/sulfide}$ = 4, $D_{Cu}^{MSS/sulfide}$ = 0.2 [2,19,79].

As demonstrated above, the disseminated sulfide in the Jinchuan deposit did not experience the distinct fractional crystallization of sulfide liquid. Thus, we took the average $Cu_{sul}$ and $PGE_{sul}$ tenors of disseminated sulfide in the three parts of the Jinchuan deposit as the initial values of liquid sulfide fractionation, respectively. After the statistical analysis of all data from the Jinchuan deposit,

we obtained an initial sulfide melt composition of 139.3 ppb $Rh_{sul}$ and 7.69% $Cu_{sul}$ (n = 63) for the western intrusion, 63.4 ppb $Rh_{sul}$ and 6.07% $Cu_{sul}$ (n = 20) for segment II-W and 45.4 ppb $Rh_{sul}$ and 6.82% $Cu_{sul}$ (n = 19) for segment II-E. The plots of $Rh_{sul}$ and $Cu_{sul}$ in net-textured, massive and Cu-rich sulfide, and the modeling of sulfide fractionation are illustrated in Figure 13.

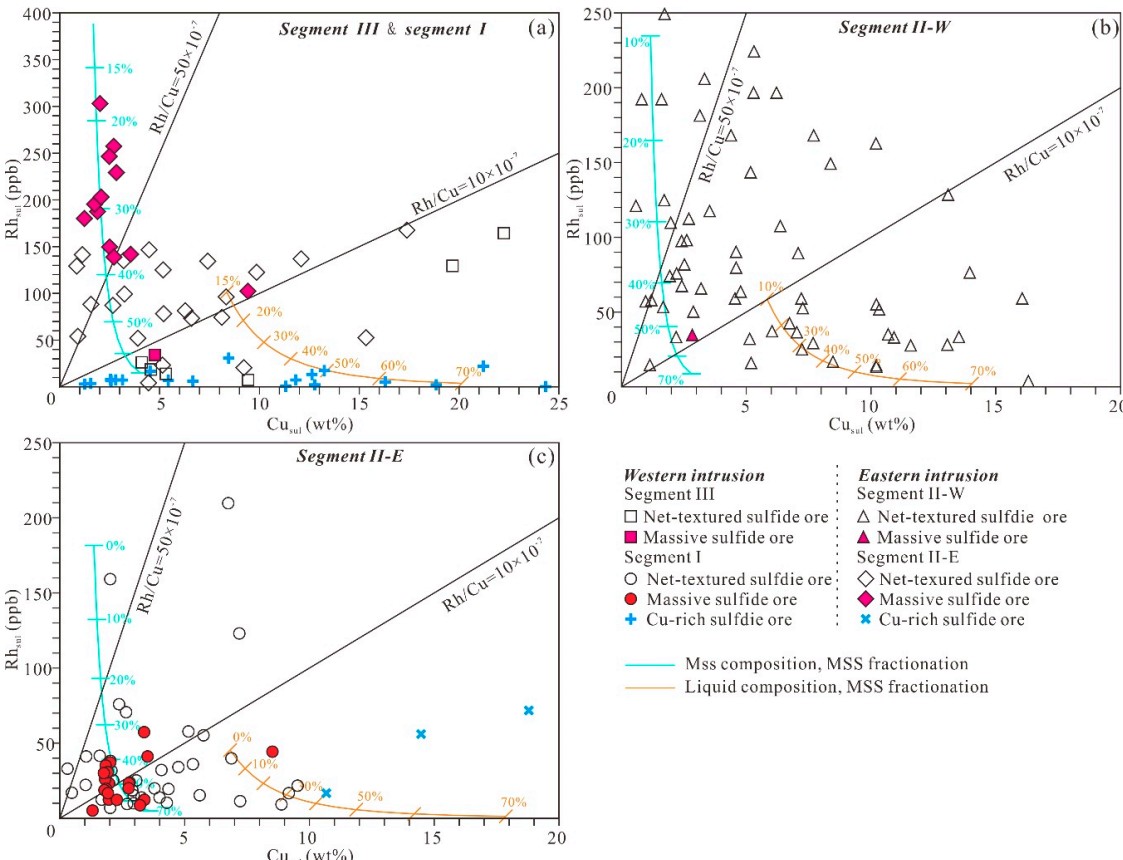

**Figure 13.** Plot of ppb Rh versus wt.% Cu in 100% sulfides for net-textured and massive sulfide ores from the Jinchuan deposit. The data are from this study and [13,16,17,29,33,36].

The net-textured sulfide in the western intrusion and segment II-E shows a positive relationship between Rh and Cu (Figure 13a,c), illustrating the remarkable effect of R-factors [56]. Our observation of no correlation between $Rh_{sul}$ and $Cu_{sul}$ in segment II-W (Figure 13b), together with the negative correlation between IPGE and PPGE (Figure 7e,f), supports the early fractional crystallization of MSS in a staging magma chamber or magma conduit. Therefore, we suggest that the formation of net-textured sulfide in the Jinchuan intrusion was both controlled both by the R-factor and sulfide fractionation. Our modeling shows that the massive ore in segment I was formed by ~20% to 40% fractionation of the sulfide melt. All samples of Cu-rich sulfide in the eastern part of segment I have extremely low IPGE tenors, corresponding to residual liquid sulfide after the fractionation of sulfide melt (Figure 13a). On the other hand, the massive ore in segment II-E was formed by ~40% to 60% fractionation of the sulfide melt (Figure 13c). The $Rh_{sul}/Cu_{sul}$ ratios of massive ore in segment II-E are much lower than segment I, which resulted from more highly differentiated sulfide liquids [2]. This is consistent with the estimated result of a higher degree of sulfide segregation for segment II-E, as estimated by the Cu/Pd ratios (Figure 11).

The compositional variations of massive sulfide within a magmatic sulfide deposit, in terms of fractional crystallization, can be further explained by a plot of Rh/Cu versus Rh [2,56,78]. The initial tenors in the sulfide liquid and the partition coefficient between MSS and sulfide liquid for Rh and Cu are the same as before. The massive and Cu-rich sulfides in segments I and II-E are mostly contained

and scatter along the lines for liquid fractionation and MSS (Figure 14). The massive sulfide in segment I show a narrow variation of Rh$_{sul}$, and Cu-rich sulfide are further determined to be formed by the ~50% to 70% fractionation of sulfide melt (Figure 14a). However, the massive sulfide samples from segment II-E show a positive correlation between Rh/Cu and Rh and scatter along the line of sulfide liquid and MSS fractionation (Figure 14b). This feature is quite similar to the Ovoid dyke in the Voisey's Bay deposit, which is close to the entrance of the shallow magma chamber [2,56]. As proposed by Naldrett et al. [56], nearly straight vertical trends on plots of Rh/Cu versus Rh of massive sulfides are the result of the modification of the composition by the redistribution of Cu, subsequent to the original crystallization. Positive correlations suggest that the orebody crystallized from the base upwards, with fractionated liquid becoming concentrated within a lens near the top of the orebody. Considering that the massive sulfide has a narrow variation of Rh$_{sul}$ and lower fractionation than the Cu-rich sulfide in segment I, we suggest that the Cu-rich sulfide was formed by the redistribution of Cu following the solidification of the sulfide melt. The massive sulfide in segment II-E was formed by a ~40% to 60% fractionation of sulfide melt and the orebody crystallized from the base upward. The lens near the top of the body is commonly found in segment II-E (Figure 2c), which provides additional support for the above interpretation.

*6.4. Multiple Magma Conduits Model for the Jinchuan Deposit*

The formation of the Jinchuan deposit has long been explained by a magma conduit system as a whole. However, two individual intrusions comprised of the Jinchuan deposit have been recognized not only by stratigraphic sequences, the spatial distribution of rock and mineralization types [13,36], and quartz syenite (Figure 2a), but also by the characteristics of PGEs (Figures 6 and 8), PCA and PLS-DA (Figure 9). Our model suggests that three parts (western intrusion, segment II-W and segment II-E) of the Jinchuan sulfides were formed by PGE-depleted magma with different amounts of prior sulfide removal. Additionally, the variation of Pt/Pd and the (Pt + Pd)/(Ir + Ru + Rh) ratios along the strike in three parts of the Jinchuan deposit show different trends and are in accordance with the R-factor. Thus, the differential depletion of PGEs and the internal variation of Pt/Pd and the (Pt + Pd)/(Ir + Ru + Rh) ratios indicates three parts of the Jinchuan deposit were most probably formed by three individual magma conduits.

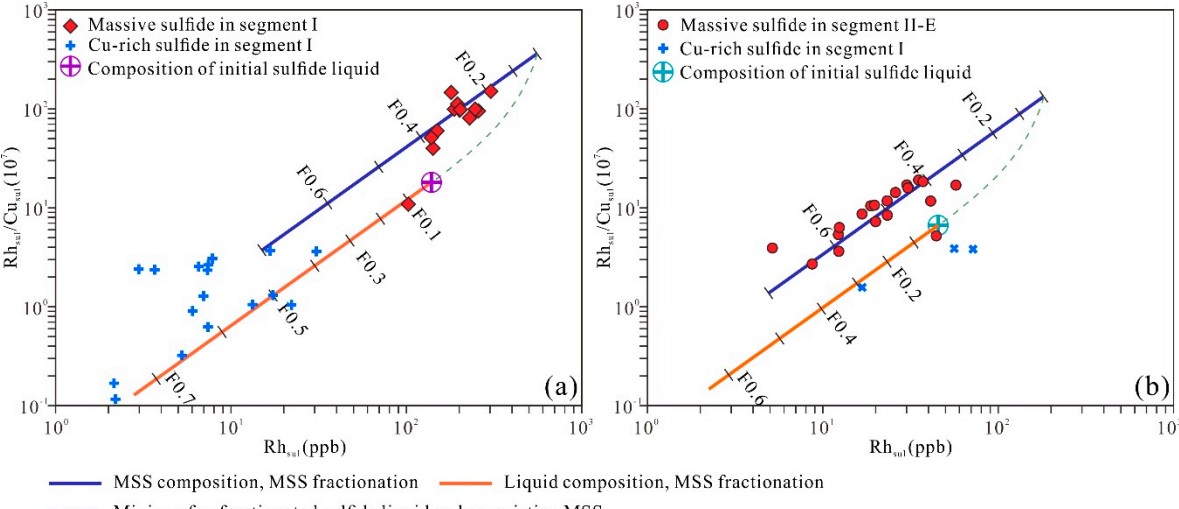

**Figure 14.** Plots of Rh/Cu versus Rh in 100% sulfides for the massive and Cu-rich sulfides in the western intrusion (**a**) and segment II-E (**b**), showing the envelope defined by model curves for perfect Rayleigh fractionation of the sulfide liquid, the monosulfide solid solution in equilibrium with this liquid and mixtures of varying proportions of the unfractionated liquid and coexisting monosulfide solid solution. The data are from this study and [13,16,17,29,33,36].

In the preceding discussion, we have proposed a model for the genesis of sulfide ores involving multiple magma conduits in the Jinchuan intrusion (Figure 15). In this model, PGE-undepleted and PGE-depleted magmas ascended along three magma conduits and intruded at Jinchuan sub-vertical normal fault systems. Additionally, the internal variation of Pt/Pd, the (Pt + Pd)/(Ir + Ru + Rh) ratios and the R-factor are explained by metal upgrading by PGE-depleted magma flow [16,17]. After the emplacement of PGE-undepleted magma, which was generated in the dunite and lherzolite, with no or weakly disseminated sulfide, PGE-depleted magma intruded at Jinchuan through three different conduits and formed the Jinchuan sulfide ores (Figure 15a). For the western intrusion, the PGE-depleted magma was locally emplaced deep in the eastern zone of segment III and moved toward both sides of the western intrusion. The massive sulfide in segment I was formed by the ~20% to 40% fractionation of sulfide melt. The Cu-rich sulfide was formed by the ~50% to 70% fractionation of sulfide melt and Cu was redistributed during the solidification of the sulfide melt. For segment II-W, the PGE-depleted magma was locally emplaced in the deep center area and moved toward the western and eastern sides. The negative correlation between IPGE and PPGE and no correlation between Rh and Cu suggests a minor amount of MSS fractionation in the staging chamber or magma conduit. For segment II-E, the PGE-depleted magma was most likely emplaced in the deep part of the middle portion. The massive sulfides were formed by ~40% to 60% fractionation of the sulfide melt, which settled down under the action of gravity. The unvarying metal ratios (Pt/Pd and (Pt + Pd)/(Ir + Ru + Rh)) and the R-factor in segment II-E may be caused by the settling of sulfide melt under gravity or the emplacement of magma in fractures, or both. Furthermore, the crystallization of the sulfide from the base upward in segment II-E is consistent with the upward increasing Ni content in the original sulfide liquids [37].

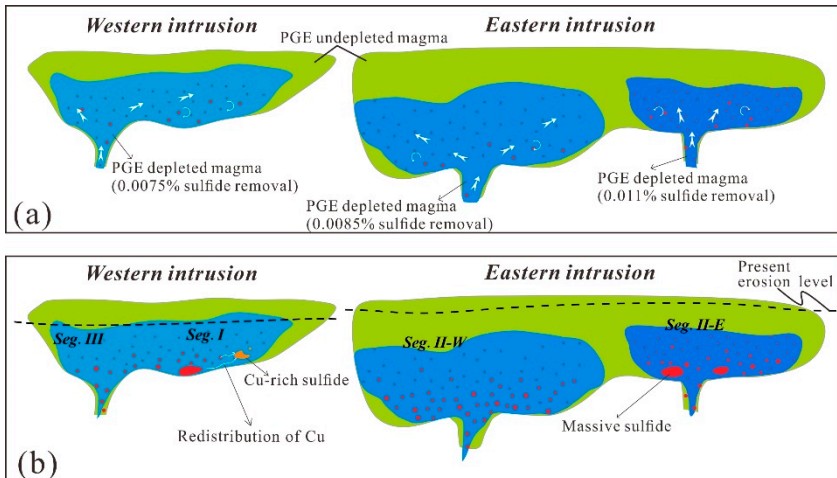

**Figure 15.** Multiple magma conduits model for the Jinchuan deposit. (**a**) PGE-depleted magmas with different amounts of depletion were emplaced through three independent magma conduits. (**b**) MSS was fractionated and gathered to form the massive sulfide, and the Cu-rich sulfide in segment I resulted from the redistribution of Cu.

## 7. Conclusions

(1) The parental magma of the Jinchuan intrusion was made up of high Mg picritic basalts, which were depleted in PGEs due to sulfide removal (0.0075% prior sulfide removal for the western intrusion, 0.0085% for segment II-W and 0.011% for segment II-E) from picritic basalt (100 ppm Cu, 1 ppb Ir and 10 ppb Pd) before emplacement.

(2) The disseminated and net-textured sulfides from the Jinchuan deposit were formed by sulfide segregation from PGE-depleted magma under a similar range of R-factors (100 to 50,000), where the metal ratios (Pt/Pd and (Pt + Pd)/(Ir + Ru + Rh)) decrease towards both sides of $F_8$ in the western intrusion and the center in segment II-W, but show no lateral variation in segment II-E.

(3) The massive sulfides in segments I and II-E were formed by the ~20% to 40% and ~40% to 60% fractionation of sulfide melt, respectively, and the latter crystallized from the base upward.

(4) Three individual magma conduits (east of segment III, center of segment II-W and beneath segment II-E) contributed to the formation of sulfide ores in the western intrusion and segments II-W and II-E, respectively.

**Supplementary Materials:** The following are available online at http://www.mdpi.com/2075-163X/9/3/187/s1, Table S1: Major element (%) and trace element (ppm) compositions of the samples from the Jinchuan deposit.

**Author Contributions:** X.M., L.L., and R.Z. conceived and designed the experiments; Z.L., R.Z., J.M.D., B.Y., and Q.A. took part in the discussion; X.M., L.L., Z.L., and J.M.D. wrote the paper.

**Funding:** This work was jointly funded by projects from the National Key R&D Program of China (No. 2017YFC0601503), the National Natural Science Foundation of China (No. 41772349), the Project of Innovation-driven Plan of Central South University (No. 2018zzts689), the Jiangxi Provincial Department of Education (Grant No. GJJ170463), and the Open Research Fund Program of Key Laboratory of Metallogenic Prediction of Nonferrous Metals and Geological Environment Monitoring (Central South University), Ministry of Education (Grant No.2018YSJS05).

**Acknowledgments:** We thank Yuanzhi Meng and Dexian Li of the Nickel Cobalt Research and Design Institute, Jinchuan Group Co., Ltd. for sharing their knowledge about the local geology and their support during field work. Wenzhou Xiao, Qiujiao He, Ye Liu and Rijun Du of the Geoscience and Info-physics of Central South University, China, were very capable field assistants. Comments by the Editor and two anonymous reviewers helped improve the manuscript substantially, and they are gratefully acknowledged.

**Conflicts of Interest:** The authors declare no conflict of interest.

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
