# Peer review of "Multiple Magma Conduits Model of the Jinchuan Ni-Cu-(PGE) Deposit, Northwestern China: Constraints from the Geochemistry of Platinum-Group Elements"

_minerals, doi:10.3390/min9030187_

Round 1

Reviewer 1 Report

Good work! Nothing to recommend except few typing errors, mostly in the references 

Author Response

Dear Reviewer,

 Thanks for your support to our study.

All of the marked texts have been modified in the revised manuscript. And we have also conducted several rounds of proofreading and fixed all the typos and grammar errors that we can find.

Yours sincerely,

Xiancheng Mao

Longjiao Li

Zhankun Liu (zkliu0322@csu.edu.cn)

Renyu Zeng

Jeffrey M. Dick

Bin Yue

Qixing Ai

Reviewer 2 Report

The manuscript is devoted to one of the largest Ni deposits in the world,  the Jinchuan deposit. The main idea of the authors is that Intrusive bodies in which ores are localized are formed as a result of the emplacement of different portions of magma. Such an assumption may indeed reflect the reality.

To prove this assumption, the authors use two types of data: 1) the composition of Intrusive rocks, 2) the composition of ores, namely the PGE ratios for the R-factor determination . These data have different meanings for the solution of the problem. Data on the composition of rocks are more important for comparing the initial melts. However, their presentation raises some questions. Given spider-diagrams do not show clearly the differences in rock compositions. It is necessary to give a number of diagrams for different Intrusive bodies, which will clearly show that the rock compositions occupy separate fields within its. The authors should choose the elements that reflect this difference. As a rule, the difference should be most evident in the ratios Gd/Yb, La/Sm, La / Yb. However, the given Cu/Ni ratio is not very informative because  these elements are poorly defined by ICP-MS. In the case of the use of these metals, the authors should show the definition error for each element  in the Supplementary  table: it is necessary to give the detection limit for each element, as well as errors in their definition on the basis of standards.   Very good data must be used  as well as adequate their presentation to solve the problem of different intrusions. For example, the Ta/Nb ratio is different for rocks on Figure 5( g). It looks as a mistake.

The use of different ratios of metals in ores seems generally unpromising for the solution of this problem. Variations in metal ratios within the orebodies are explained, as a rule, by the fractionation of the sulfide melt itself. This is well known for the main orebody of the Oktyabr’koe deposit in the Norilsk region, where sulfide ores have a very complex zoning  with different ratios of Pt/Pd, Pt+Pd/Rh+Ir+Ru.  The same conclusion can be done from the data of the authors of the article, which show wide range of metals variations  within a single ore body, which can hardly be explained by different R - factor. There are no real laws for metal variations . If the model trends are compared with the real trends , the correlation coefficients should be given for the ore trends.

There are  small mistakes as well. For example, the authors refer to the work of Naldrett et al., 1992, in which the Tuklonsky basalts are considered as a primary magma. However, there are many publications where this assumption was rejected. Furthermore, it was shown the absence of the plumbing system in the Norilsk area. The authors have to discuss these data, if they give data on the Tuklonsky Formation  for comparison with the Jinchuan.  In addition, The authors on Figure 8 (f) give patterns for disseminated and  massive ores from the Norilsk. But disseminated  ores  of different deposits (Norilsk 1, Talnakh and so on) have different composition, very often theier composition varies  within single deposit.  It is necessary to specify compositions of which ore are given.

Author Response

Response to Reviewer 2 Comments

Point 1: The manuscript is devoted to one of the largest Ni deposits in the world, the Jinchuan deposit. The main idea of the authors is that Intrusive bodies in which ores are localized are formed as a result of the emplacement of different portions of magma. Such an assumption may indeed reflect the reality.

To prove this assumption, the authors use two types of data: 1) the composition of Intrusive rocks, 2) the composition of ores, namely the PGE ratios for the R-factor determination. These data have different meanings for the solution of the problem. Data on the composition of rocks are more important for comparing the initial melts. However, their presentation raises some questions. Given spider-diagrams do not show clearly the differences in rock compositions. It is necessary to give a number of diagrams for different Intrusive bodies, which will clearly show that the rock compositions occupy separate fields within its. The authors should choose the elements that reflect this difference. As a rule, the difference should be most evident in the ratios Gd/Yb, La/Sm, La / Yb. However, the given Cu/Ni ratio is not very informative because these elements are poorly defined by ICP-MS. In the case of the use of these metals, the authors should show the definition error for each element in the Supplementary table: it is necessary to give the detection limit for each element, as well as errors in their definition on the basis of standards. Very good data must be used as well as adequate their presentation to solve the problem of different intrusions. For example, the Ta/Nb ratio is different for rocks on Figure 5(g). It looks as a mistake.

Response 1: First of all, we would like to thank you for the professional comments on our manuscript. To better to prove the fact of two individual intrusions at Jinchuan, we have followed your suggestions to distinguish the two intrusions by geochemical features, including La/Sm, Gd/Yb and La/Yb ratios and statistical methods (such as principal component analysis (PCA) and Partial least squares-discriminant analysis (PLS-DA)) that are popular to verify classification and identify the difference between different datasets. Although we try our best to do a series of diagrams to reflect the difference in the two intrusions, these diagrams did not show very clear geochemical regularity, and only the La/Sm ratio of the eastern intrusion (2–4.7) show narrower range than western intrusion (0.6–6.3). Nevertheless, the PCA and PLS-DA diagrams (Figure 9) could clearly distinguish the two intrusions, especially seeing the significant large PC1 (close to 80%). Thus, in the revised manuscript, our geochemical results have overall reflected the differences in rock compositions of the western and eastern intrusions at Jinchuan.

Besides, we have carefully checked the reliability of geochemistry data. The analysis for the Cu and Ni was analyzed by Atomic Absorption Spectroscopy (AAS) rather than ICP-MS. The mistake has been corrected in the revised manuscript. The detection limit and relative standard deviation (RSD) of Cu, Ni, S, and PGE have been listed in Table 1. The analysis results are significantly higher than the detected limit. Thus, the geochemical data in our manuscript are reliable.

We also added the detection limit for all major and trace elements, as well as errors in their definition on the basis of standards in the supplementary table. In the revised manuscript, all the data below or near detection limit (e.g., U and Ta) were not used to discuss any issues and plotting. Due to the very low content of Ta, the element has been removed in spider-diagrams (Figure 5) and others.

Point 2: The use of different ratios of metals in ores seems generally unpromising for the solution of this problem. Variations in metal ratios within the orebodies are explained, as a rule, by the fractionation of the sulfide melt itself. This is well known for the main orebody of the Oktyabr’koe deposit in the Norilsk region, where sulfide ores have a very complex zoning with different ratios of Pt/Pd, Pt+Pd/Rh+Ir+Ru. The same conclusion can be done from the data of the authors of the article, which show wide range of metals variations within a single ore body, which can hardly be explained by different R - factor. There are no real laws for metal variations. If the model trends are compared with the real trends, the correlation coefficients should be given for the ore trends.

Response 2: Thank you. We have added the analysis of various ratios of Pt/Pd and (Pt+Pd)/(Rh+Ir+Ru) of sulfide ores after reading the papers and books about Norilsk. In revised manuscript, it is clearly found that both of the two ratios are decrease toward both sides of the line III-5 in the western intrusion and decrease toward both sides of the line II-14, whereas no significant spatial variation was observed in the segment II-E. These variation trends, controlled by the fractionation of sulfide melt, are most likely reflected the magma flow from the center to margin. It is needed to point out that not only the metal ratios but also R-factor in segment II-E show no lateral variation. It likely results from the unobvious lateral magma flow after magma emplacement in segment II-E. Thus, the variation of Pt/Pd and (Pt+Pd)/(Rh+Ir+Ru) in three parts powerfully supported the multiple magma conduits model for the Jinchuan deposit. It is also mentioned that the spatial variation of R-factor at Jinchuan shows a generally similar trend to these ratios. However, it is not clear the association and a possible explanation is that the calculation of R-factor is based on the analysis results of PGE. As you said, a wide range of metals variations within a single ore body is hardly be explained by different R-factors. Thus, to make a more rigorous discussion, we have removed the discussion about R-factor controls on a single orebody. In fact, most discussions of the paper mainly are rarely associated with the evolution of a single orebody and the major contribution of this study is the new understanding of ore genesis by the multiple magma conduits model. Thus, the revise did not affect the final contribution, on the country, confirming it by the ratios of Pt/Pd and (Pt+Pd)/(Rh+Ir+Ru). Here, we would like to thank you again for the constructive suggestions.

Point 3: There are small mistakes as well. For example, the authors refer to the work of Naldrett et al., 1992, in which the Tuklonsky basalts are considered as a primary magma. However, there are many publications where this assumption was rejected. Furthermore, it was shown the absence of the plumbing system in the Norilsk area. The authors have to discuss these data, if they give data on the Tuklonsky Formation for comparison with the Jinchuan. In addition, the authors on Figure 8 (f) give patterns for disseminated and massive ores from the Norilsk. But disseminated ores of different deposits (Norilsk 1, Talnakh and so on) have different composition, very often their composition varies within single deposit. It is necessary to specify compositions of which ore are given.

Response 3: For these mistakes, we have carefully corrected in the revised manuscript, after reading the relevant literature. Considering the controversy of ore genesis and primary magma at Noril'sk region, we deleted the comparison of magmatic plumbing system with Norilsk area in the introduction and the assumption of a primary magma of the Tuklonsky basalts in section “6.1. Prior Sulfide Segregation”. Because the magma conduit model in Noril'sk is rejected, the comparison of disseminated and massive sulfide in Figure 8 was deleted. We have added the specified deposit of which ore are given in Voisey's Bay.

Overall, we believe that after the revise, the quality of our manuscript has been greatly improved. Please let us know if any further changes are needed and we will be very happy to revise it and make it publishable.

Yours sincerely,

Xiancheng Mao

Longjiao Li

Zhankun Liu (zkliu0322@csu.edu.cn)

Renyu Zeng

Jeffrey M. Dick

Bin Yue

Qixing Ai

Round 2

Reviewer 2 Report

The authors took into account almost all the comments of the reviewer and tried to make more clearly the differences between the intrusions formed by different magmas. In fact, the problem of definition of separate magmas impulses is very complex. These different portions of magma could have very close (or even identical) compositions. This conclusion follows from the authors’ data. This article makes a certain contribution to the solution of this problem. In general, of course, the origin of the deposit is still under examination.   I hope that the authors will work on it in the future, developing new approaches and using new methods for real model constructions.